# FleetAgent: Natural Language Driving Explanation and Evaluation for Vehicle Teleoperation

## Abstract

Large-scale driverless fleets rely on teleoperation to resolve rare, safety-critical edge cases that onboard autonomy cannot handle robustly. We introduce FleetAgent, a cloud-hosted multimodal large language model (MLLM) that assesses the plan and context of an autonomous vehicle (AV) to decide whether teleoperation is needed. FleetAgent consumes a compact vectorized representation of observations and planned actions rather than raw sensors, and produces a natural-language explanation and evaluation towards the traffic scenario and driving decision. A dedicated vector encoder replaces conventional text tokenizers and vision encoders, substantially reducing the number of input tokens and server memory footprint while preserving the information needed for proper functioning. We also build a dataset based on nuScenes and augment it with synthetic imperfect driving decisions and annotated explanation and evaluation labels. System-level studies indicate a maximum $625\times$ reduction in communication demand and a maximum $16.54\times$ reduction in cache size. Model-level experiments also show competitive response quality and plan-evaluation accuracy, with a 41% improvement in BLEU score and an 11% reduction in task failure rate. Because all computation runs on the cloud, the approach introduces no additional onboard burden. Together, these results outline a practical path to scalable, explainable teleoperation support for AV fleets, paving the way to another paradigm for MLLMs' application in autonomous driving.

## 1 Introduction

Autonomous driving technology has been evolving rapidly over the last decade, with companies like Waymo, Zoox, and Tesla starting to deploy driverless fleets in major U.S. cities to the general public Waymo (2025); Zoox (2025); Tesla (2025). Large-scale driverless fleets sometimes rely on remote operation as a backup to handle rare and safety-critical situations that the onboard system fails to resolve. Specifically, remote operation can take the forms of remote monitoring, direct control, or human guidance, which have already been used by leading robotaxi operators.

Typical remote driving systems feature a bi-directional communication, where an autonomous vehicle's (AV) sensor data, such as camera and LiDAR feeds, is transmitted over a wireless network to a human operator at a remote center. The operator assesses the situation and sends guidance back to the vehicle. To mitigate the safety risks associated with network latency, some industry leaders are adopting a new paradigm where the AV remains in full control of its low-level operations while the human provides high-level, strategic guidance. However, vehicle-to-server sensor streaming remains challenging in terms of latency and cost as the system scales up.

Moreover, with the advancement of vehicular automation, the lower intervention rate prompts the need for an approach that can accurately trigger teleoperation. These realities motivate the need for an intelligent system that **continuously evaluates** vehicle behavior and, at the right moments, reminds human operators and provides **intuitive explanation** of the situation with **minimum volume of message** transmitted.

Currently, multi-modal large language models (MLLMs) have shown great potential as generalist reasoners for various complex tasks, including autonomous driving. This makes them ideal for

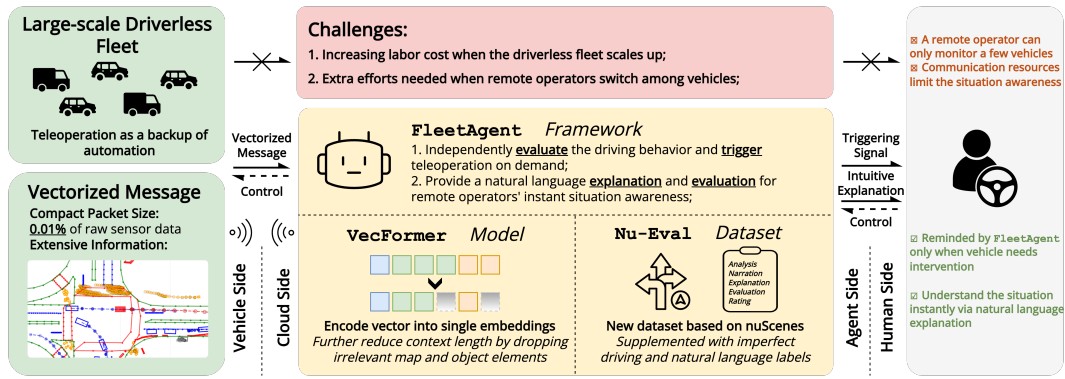

Figure 1: FleetAgent: A Framework for Large-Scale Driverless Fleet Teleoperation.

serving as standalone agents to identify challenging scenarios and wrong driving decisions from the massive amount of vehicles in the fleet. It also natively provides the remote operator with an intuitive explanation of what is happening to the vehicle that requires intervention. While existing research mainly focuses on grounding MLLMs in the driving domain from different stages, such as camera perception, motion prediction, and vehicle planning, we aim to start from the actual limitations of deploying MLLMs for real-world applications and explore a new way of MLLM usage in autonomous driving.

In this paper, we present an LLM-based system for driving explanation and evaluation tailored for remote driving operation via vehicle-to-network (V2N) communication. The key contributions of this paper are as follows:

1. We propose FLEETAGENT, a flexible framework based on vectorized information presentation for utilizing MLLM on a cloud server to provide 1) the evaluation of the AV driving behavior and 2) the explanation for the remote drivers' quick situation awareness. FLEE-TAGENT works as a connector between the large-scale driverless fleet and human teleoperators.

2. A dataset NU-EVAL extended from nuScenes, providing imperfect driving and extensive natural language labels on analysis, explanation, and evaluation; a dedicated multi-modal encoder VECFORMER in place of general-purpose text tokenizers and pretrained visual encoders, greatly reducing the number of input tokens and hence reducing the graphics memory footprint during both training and inference.

3. Both model and system-level evaluations showcase the outstanding capability of our proposed framework and model, meeting the communication and computation requirements in real-world applications. Notably, no additional computation is needed on the AV side.

## 2 RELATED WORKS

### 2.1 REMOTE DRIVING SYSTEMS

Previous research on vehicular and robotic teleoperation mainly focus on the challenge of real-time control over communication network, where the latency effect is detailedly measured by Georg et al. (Georg et al., 2020): Neumeier et al. (Neumeier et al., 2022) proposed a multi-step approach to reduce the data rate needed for teleoperated driving; Dmitrij et al. (Schitz et al., 2021) mitigates the need of low-latency communication by combining human's high-level instruction and vehicle's low-level planning, building a shared autonomy framework; Xie et al. (Xie et al., 2021) proposed a predictive display method for robotic teleoperation, addressing the network latency challenge by presenting a near-future view of the robot; Gohar and Lee (Gohar & Lee, 2020) proposed a remote operator selection method to match suitable remote operators with fleet vehicles;

From the remote operator aspect, previous research focuses on topics like improving human-machine interfaces (HMIs) and operation quality: Wolf et al. (Wolf et al., 2024) provided an in-

depth examination of different teleoperation interfaces; Tsagkournis et al. (Tsagkournis et al., 2023) and Yang et al. (Yang & Michael, 2020) mentioned that intention prediction of robot teleoperators can assistively reduce operator workload and improve the overall performance; Cho et al. (Cho et al., 2023) provided a denoising-based method robust to imperfect input from unskilled drivers.

Current design of remote driving systems has not focused on the process of identifying whether a fleet vehicle needs intervention and a teleoperation session is requested from the vehicle side, usually via the constraints of operational design domain and techniques for edge case detection (Rahmani et al., 2024). We aim to close this gap by adopting a natural language evaluating and explaining mechanism outside the vehicle, while introducing no additional computation cost and a limited amount of transmission cost.

### 2.2 MULTIMODAL LARGE LANGUAGE MODELS (MLLMS) IN AUTONOMOUS DRIVING

The application of MLLM in the autonomous driving domain has evolved into multiple paradigms (Wang et al., 2025; Sima et al., 2025; Tian et al., 2024; Park et al., 2025), including visual question answering (VQA) and integrated end-to-end systems (Li et al., 2025; Hwang et al., 2024; Zhou et al., 2025; Xu et al., 2024), where multimodal LLMs are used for the planning of AVs. Agent-Driver(Mao et al., 2024) proposed an LLM-agent-based method to provide explainable action. To enhance the aligned interpretability, Hint-AD (Ding et al., 2024) generates language output aligned with the autonomous driving model output, and the paper provided a driving explanation dataset Nu-X. Moreover, ALN-P3 (Ma et al., 2025) proposed a distillation approach to transfer the knowledge from multimodal LLM to a light autonomous driving model. Notably, in most prior research, due to the actual needs of the VQA task and vehicle planning task, multimodal LLMs are designed to deploy onboard, and the model inputs typically include language instructions, raw sensor data, and ego state information. This information can losslessly describe what's happening around and inside the vehicle, but is too large to be transmitted over the network.

### 2.3 VEHICLE-TO-EVERYTHING COMMUNICATION

Vehicle-to-everything (V2X) is another rapidly evolving domain, built on top of several network interfaces including DSRC and C-V2X (Abboud et al., 2016). V2N connects vehicles to cloud infrastructures and is a subset of V2X, enabling network-based data exchange. While the advancement of communication technology has enabled a short enough latency for delivering teleoperation commands, the latency from image captured to image displayed remotely is still high and unstable (Testouri et al., 2025), indicating that a smaller message size is not only the requirement of cost but also a critical factor of the system performance. In the vehicle-to-vehicle and vehicle-to-infrastructure scenario with no flow control and small conflict probability, reducing transmitted message size can accelerate transmission by more than five times (Zhao et al., 2025). In a busier vehicle-to-network scenario, the gain from reducing message size would be more significant.

## 3 PROBLEM FORMULATION

We formulate the driving explanation and evaluation task for remote teleoperation, which differs fundamentally from existing VQA tasks and ego vehicle planning tasks. Traditional VQA systems in autonomous driving follow the paradigm: $f(I, Q) \to A$, where raw sensor input $I$ and questions $Q$ produce answers $A$. Ego vehicle end-to-end planning models map $f(I, S) \to P$ from raw sensor input $I$ and ego vehicle state $S$ to planning decisions $P$. In contrast, our task takes the form: $f(O, P) \to (L, i)$, where intermediate observation of the surrounding $O$ and the vehicle's planning output $P$ are evaluated to produce natural language evaluation and explanation $L$ and intervention score $i$.

This formulation introduces two key challenges for teleoperation: operating under V2N bandwidth constraints with compact context inputs, and evaluating the appropriateness of planning outputs rather than directly working on the planning process. The pipeline addresses these by generating a natural language explanation of the traffic scenario and driving decision, evaluating that decision, and providing a quantized intervention score. These outputs provide situational awareness and structured guidance for operators while limiting the need for extra computation and communication costs.

# 4 DATASET

In this section, we will analyze the dataset requirements for our driving explanation and evaluation tasks, as described in Sec. 3, and introduce the annotation process for the dataset.

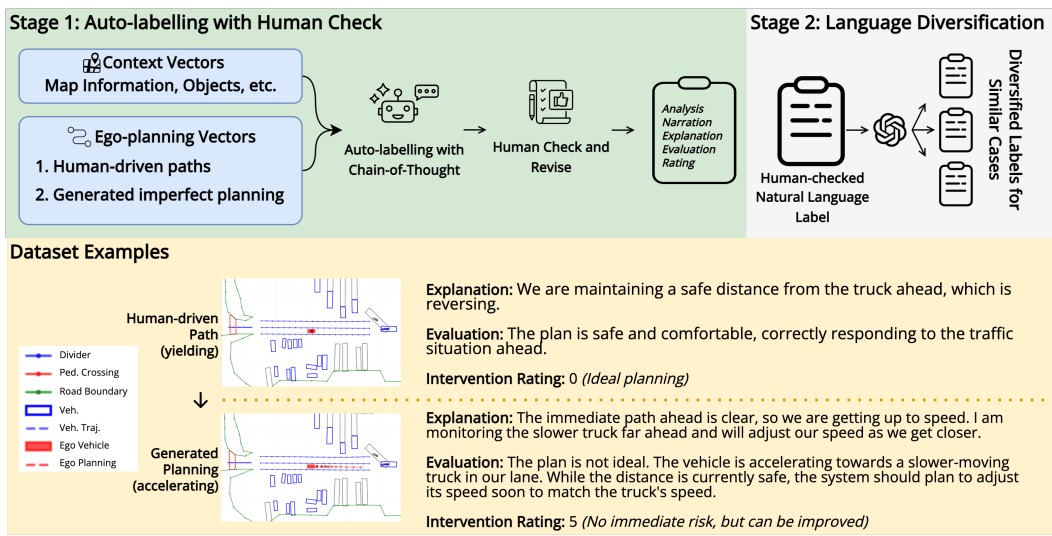

Figure 2: Annotation Pipeline and Dataset Examples.

## 4.1 TASK REQUIREMENT AND EXISTING DATASET COMPARISON

Our task involves explaining and evaluating a given ego-vehicle planning $P$ in a specific context from observations $O$ (e.g., lanes, pedestrians, vehicles). The model must produce a scene-grounded natural language response $L$ and a graded intervention urgency $i$. For learning fine evaluation and judgement rather than mere narration, training instances should include multiple planning variants, cover rare and unsafe scenarios, and events.

Previous datasets in autonomous driving, including nuScenes (Caesar et al., 2020), Waymo Open (Sun et al., 2020), KITTI (Geiger et al., 2013), and nuPlan (Caesar et al., 2022), offer rich sensor logs covering diverse traffic scenarios and high-quality human-driven trajectories. These provide the contexts $O$ and the ideal portion of candidate plannings $P$ we need, while still missing language annotations and the imperfect portion of candidate plannings.

Multiple VQA extension datasets based on nuScenes and nuPlan, such as NuScenes-QA (Qian et al., 2024), NuInstruct (Ding et al.), NuPrompt (Wu et al., 2025), NuPlanQA (Park et al., 2025), and DriveLM (Sima et al., 2025), primarily target captioning, grounding, and scene understanding rather than evaluating and explaining a given plan, which makes them not directly suitable for our task. For existing driving explanation and reasoning datasets, including Nu-X (Ding et al., 2024) and BDD-X (Kim et al., 2018), annotations center on human driver actions, which are targeted for interpretability studies. However, those datasets neither provide evaluation on the candidate planning $P$ from human drivers nor provide adequate imperfect driving planning. As identifying inefficient or unsafe driving is the main goal of our task, we build a new dataset, VECEVAL, providing a vectorized representation of each scenario and the corresponding explanation, evaluation, and the intervention urgency score. Considering the potential need for cross-validation and joint training, we base our dataset on nuScenes, one of the most popular datasets in the autonomous driving domain.

## 4.2 DATASET CONSTRUCTION PIPELINE

As shown in Fig. 2, our dataset annotation generally consists of two parts: the first part is the generation of imperfect driving behaviors, and the other part is the annotation of natural language explanation, evaluation, and intervention urgency scoring based on the human-driven ego planning and the generated imperfect driving.

### 4.2.1 Imperfect Planning Generation

In a real-world driving scenario, an autonomous system might generate incorrect planning trajectories, which are unsafe or inefficient. For those planned trajectories that are distorted or physically infeasible, we assume they can be identified through limitations on position, speed, or acceleration range. To better imitate those imperfect plans due to wrong decision making, we adopt trajectories sampled from a real trajectory vocabulary as the imperfect trajectories. To construct the trajectory vocabulary, all the human-driven trajectories from the base dataset are collected and clustered into 15 categories. The clustering successfully categorizes different trajectories regarding the speed profile (steady speed, acceleration, or deceleration) and steering status (going straight, turning left, or turning right). For each scenario, an imperfect driving trajectory is generated based on the human-driven trajectory. A category different from the human-driven trajectory is randomly allocated, and then imperfect planning is randomly sampled from that category.

### 4.2.2 Natural Language Label Annotation

After obtaining both human-driven and imperfect trajectories, we follow the steps outlined in Fig. 2 to annotate the natural language explanations, evaluations, and intervention urgency scores. In the first stage, we utilize an advanced reasoning model and chain-of-thought (CoT) prompting to generate the draft annotations from ground-truth data in the base dataset nuScenes. The CoT includes multiple steps: 1) Accessing the road and lane structure; 2) Identifying moving and static objects; 3) Describing the given planned trajectory; 4) Analyzing the interactions between the ego vehicle and surrounding objects; 5) Giving the desired explanation, evaluation, and intervention score. Human annotators will then check and refine the natural language labels. In the second stage, we follow the practice presented in HintAD (Ding et al., 2024) and diversify the annotated sample into more diverse expressions, satisfying the potential need for tuning larger-scale models and more natural human-machine interaction.

### 4.3 Examples and Key Statistics

We select human-driven samples based on their relative positions in a scenario, where 11,510 out of 28,130 and 1,693 out of 6,019 of the original annotated samples in training and validating sets are annotated, respectively. The final average interval between annotated samples in the datasets is 4.43m, ensuring a good coverage of the base dataset nuScenes while keeping the diversity of the labels. For the imperfect portion, a fixed number of two samples from a scenario is randomly selected and annotated. In total, 1,2910 annotated samples are included in the training set, and 1,993 samples are included in the validation set. The lower part of Fig. 2 shows two examples from our annotated datasets, providing an intuitive understanding of the components of our datasets.

## 5 Methodology

### 5.1 Overall architecture

As shown in Fig. 3, our method leverages the vectorized information received from remote vehicles to provide a response and signal to the remote operator, working as a standalone agent monitoring the driving behavior. Unlike previous VQA works utilizing images or videos as input, we use vectorized data to achieve a tradeoff between the transmitted information and communication costs. Road structures, objects with their predicted future locations, and ego planning can all be represented by vectors.

After the cloud server receives the messages, a straightforward approach to feed them into the LLM is to convert those vectors into a text description. Though this approach requires no additional module, the converted text descriptions incur an extremely long context length of more than 10,000 tokens per frame. Long context length then significantly increases the memory footprint and overall throughput. The situation gets worse when there are more road users and a complex road structure, as more vectors are needed to represent the large number of elements, while these cases are more likely to need teleoperation. To address this issue, we design a VECFORMER to encode the received vectorized information into embeddings.

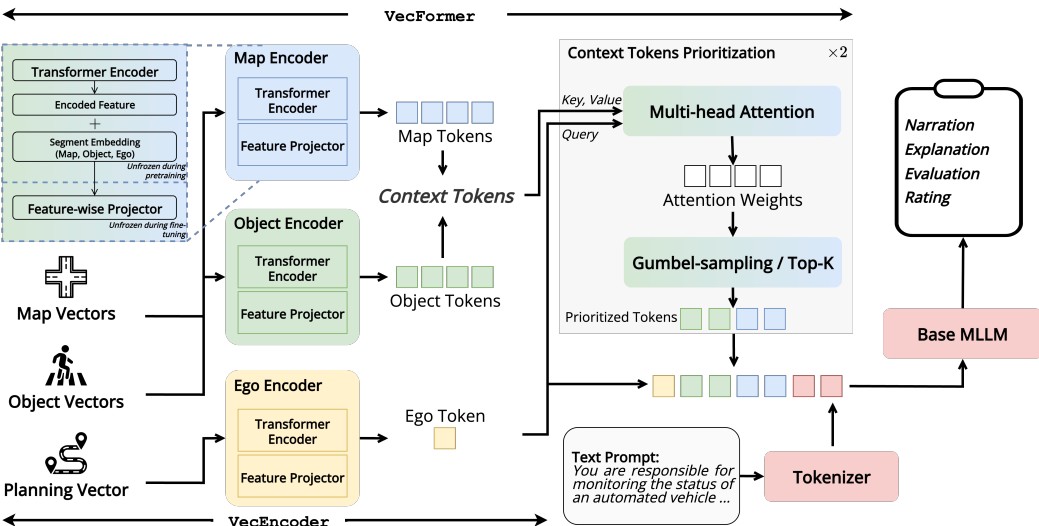

Figure 3: Underlied model design of FLEETAGENT.

We choose Qwen2.5-VL (Bai et al., 2025) as the base model because it adopts a simple but effective way of multi-modal fusion, where both textual information and multi-modal information are represented via input embeddings.

## 5.2 VECFORMER DESIGN

As shown in Fig. 3, our proposed VECFORMER consists of two parts: independent transformer encoders encoding input vectors into high-dimensional embeddings, and a multi-head gated attention to prioritize the top-K most important map and object embeddings, regarding the ego planning embedding. Denote $\mathcal{M}_i$ and $\mathcal{F}_{map,i}$ as the $i$-th vector and embedding representing a map element, such as a lane divider, a road boundary, or a pedestrian crossing zone; denote $\mathcal{O}_j$ and $\mathcal{F}_{obj,i}$ as the $j$-th vector and embedding representing an object and its predicted future trajectory; denote $\mathcal{P}$ and $\mathcal{F}_{ego}$ as the vector representing the ego-vehicle planned waypoints. Three separate transformer encoders first encode the input vectors into embeddings:

$$\mathcal{F}_{map,i} = Encoder_{map}(\mathcal{M}_i); \ \mathcal{F}_{obj,i} = Encoder_{obj}(\mathcal{O}_j); \ \mathcal{F}_{ego} = Encoder_{ego}(\mathcal{P});$$

Following the encoding stage, we employ a differentiable top-K selection mechanism using Gumbel-Softmax (Jang et al., 2017) sampling to identify the most relevant context embeddings for ego-vehicle planning. Given the ego embedding $\mathcal{F}_{ego} \in \mathbb{R}^d$ and context embeddings $\mathcal{F}_{ctx} = \{\mathcal{F}_{map}, \mathcal{F}_{obj}\} \in \mathbb{R}^{n \times d}$ where $n$ is the total number of context elements, we first compute multi-head attention scores:

$$\alpha_i = \frac{(W_Q \mathcal{F}_{ego})^T (W_K \mathcal{F}_{ctx,i})}{\sqrt{d_{head}}}$$

where $W_Q, W_K \in \mathbb{R}^{d_{head} \times d}$ are learned projection, $d_{head}$ is a hypeparameter, and $d$ is determined by the LLM input embedding dimension. To select exactly $K$ discrete context vectors while maintaining differentiability, we perform sequential Gumbel-Softmax sampling. For each selection step $k \in \{1, ..., K\}$, we sample from the categorical distribution over remaining contexts:

$$y_{soft,i}^{(k)} = \frac{\exp((\log(\pi_i) + g_i)/\tau)}{\sum_{j \in \mathcal{U}^{(k)}} \exp((\log(\pi_j) + g_j)/\tau)}$$

where $\pi_i = \text{softmax}(\alpha_i)$, $g_i = -\log(-\log(u_i))$ with $u_i \sim \text{Uniform}(0,1)$ is the Gumbel noise, $\tau$ is the temperature parameter, and $\mathcal{U}^{(k)}$ denotes the set of unselected indices. To achieve discrete selection while maintaining gradient flow, we compute: $y_{hard}^{(k)} = \text{Onehot}(\arg\max_i y_{soft,i}^{(k)})$ and $y^{(k)} = y_{hard}^{(k)} - \text{detach}(y_{soft}^{(k)}) + y_{soft}^{(k)}$. This keeps the boolean selection while allowing gradients

to flow exclusively through $y_{soft}^{(k)}$ during training. This enables the network to learn which contexts are most relevant while maintaining discrete selections, which is essential as the effect of a non-boolean selection matrix is offset by the normalization steps in the downstream LLM. The selected context embedding is then $\mathcal{F}_{sel}^{(k)} = \sum_i y_i^{(k)} \mathcal{F}_{ctx,i}$, and the selected index is masked to prevent re-sampling in subsequent steps. This mechanism ensures that exactly $K$ distinct context vectors are chosen in a differentiable manner, maintaining their individual semantic identity for downstream LLM reasoning while enabling end-to-end gradient-based optimization.

## 5.3 TRAINING STRATEGY

To efficiently adapt the VECFORMER to existing pretrained LLMs using a limited amount of annotated natural language labels, we use a training strategy including a masked vector reconstruction self-supervised pretraining, a fixed-format instruction tuning, and a final supervised fine-tuning.

**Masked Vector Reconstruction:** In this stage, the vector encoder takes the masked vectors as input, and a temporary decoder head reconstructs the encoded feature into the original unmasked vectors. This step aims to train the transformer encoder to learn the relationship among points in a vector (e.g., a lane divider or a vehicle's predicted trajectory), instead of simply projecting point positions to the latent space. Notably, no additional labelling is needed in this stage, as vanilla labels from existing datasets like nuScenes can fulfill the data needs for this stage.

**Fixed-format Instruction Tuning:** In this stage, the vector encoder functions as the input embeddings of an LLM, along with a short prompt asking the LLM to interpret the input vector. For example, an ego-vehicle planning vector is accompanied by a prompt *What is the ego planning trajectory the input vector is representing?* By freezing most of the parameters in LLM, this stage aims to align the encoded feature space with the LLM input embeddings. Similar to the first stage, this stage requires no additional natural language annotations.

**Supervised Fine-tuning:** In this stage, we use the pretrained weights from previous stages as the initial weights. During the training, only a one-layer feature projector and the context tokens prioritization module are unfrozen. The base LLM is finetuned using prefix tuning and LoRA adapter. The motivation of this design is two-fold: 1. To reduce the computational cost of fine-tuning; 2. To primarily preserve the capability for general reasoning from the base LLM, as our task requires extensive reasoning capability via the narration, explanation, evaluation, and the intervention scoring pipeline.

## 6 EMPIRICAL RESULTS

In this section, we first present the results at the system level, showcasing how our architectural design addresses real-world challenges. We then compare our model's performance with existing methods on various metrics relevant to the task of driving explanation and evaluation for vehicle teleoperation.

### 6.1 SYSTEM LEVEL RESULTS

We conduct a comprehensive system-level evaluation to assess the computational efficiency and resource requirements of different approaches, comparing both API-based and local deployment scenarios across various input modalities. Raw images are collected from surround-view or single-view RGB cameras, which are one of the most common input modalities for MLLM's application in autonomous driving. Bird's Eye View (BEV) images provide top-down perspective representations of the driving scene. Language Descriptions, which encode scene information as discrete embeddings via a text tokenizer.
As shown in Table 1, FLEETAGENT demonstrates superior system efficiency across all metrics. While raw images and BEV images require massive data transmission (25,312.5/4218.8 kB per request), FleetAgent's vector embeddings deliver similar information with only 40.5 kB, a $625\times$ reduction in bandwidth requirements, matching the compact size of language descriptions as they are both reconstructed from vectorized messages. More importantly, FLEETAGENT achieves the fastest response time of 4.41 seconds with low variance, outperforming all other methods. When compared with language description input, FLEETAGENT requires only 1,241 MB of cache memory

Table 1: System Level Comparison. Use OpenAI GPT-4o as an example for the API category, Qwen-2.5VL-7B as an example for the local model category. Due to the hardware constraint incurred by *Language Description Input* and *Raw Images Input*, all local models are inferred on half-precision. Local response time tested on a single NVIDIA A100-40GB with a batch size of 1.

| | Model Type | Input Type | Transmitted Packet Size (kB) per request | Response Time (s) avg. (s.d.) | Memory Footprint (MB) Cache/Model |
|---|---|---|---|---|---|
| 1 | | Raw Images | 25312.5 | 5.8783 (1.5160) | - |
| 2 | API | BEV Images | 4218.8 | 6.9310 (2.7076) | - |
| 3 | | Language Description | 40.5 | 7.5245 (3.0132) | - |
| 4 | | Raw Images | 25312.5 | 12.6158 (1.0635) | 10558 / 15819 ($\times 8.51$) |
| 5 | Local | BEV Images | 4218.8 | 5.8987 (1.0530) | 2604 / 15819 ($\times 2.10$) |
| 6 | | Language Description | 40.5 | 8.6372 (5.5699) | 20522 / 15819 ($\times 16.54$) |
| **FleetAgent** | **Local** | **Vector Embeddings** | **40.5** | **4.4116 (1.0610)** | **1241 / 17081** |

compared to 20,522 MB for language descriptions (a $16.54\times$ reduction) in local deployments, as the constraints from the text tokenizer are lifted and a single vector is encoded into exactly one continuous-space embedding. The significant reduction in cache memory enables faster inference and a larger batch size, given a fixed memory allocation.

Generally speaking, using vectorized messages during communication and employing VECFORMER to bridge the input and VLM achieves the most advantages from an architectural perspective. Experiments at the model-level provide quantitative results on task performance, demonstrating that FLEETAGENT's efficiency gains do not come at the cost of model capability.

## 6.2 MODEL LEVEL RESULTS

Table 2: Benchmark Comparison on NU-EVAL dataset. FLEETAGENT outperforms other baseline methods on all metrics and performs similarly to Gemini-2.5-Flash, which is used to assist annotating during the dataset construction process. **Bold** text denotes the best performance, and underlined text indicates the second-best.

| Model | Input Modality | Language Metrics | | | Lingo-Judge | | Intervention Failure Rate (%)↓ |
|---|---|---|---|---|---|---|---|
| | | B ↑ | M ↑ | R ↑ | Acc. ↑ | Score ↑ | |
| | Raw Images | 61.65 | 24.59 | 22.34 | 18.26 | 0.2304 | 15.24 |
| GPT-4o | BEV Images | 62.90 | 24.16 | 20.92 | 15.49 | 0.2191 | 16.05 |
| | Language Description | 45.05 | 29.07 | 22.21 | 26.03 | 0.2422 | 13.63 |
| Qwen-2.5VL-7B | Raw Images | 66.28 | 26.43 | 27.58 | 55.63 | 0.3281 | 15.90 |
| (Few-shot Example) | BEV Images | 42.16 | 21.84 | 17.78 | 22.25 | 0.2533 | 13.83 |
| | Language Description | 48.38 | 29.84 | 22.21 | 52.22 | 0.3056 | 15.14 |
| FleetAgent (w/o tokens prioritization) | Vector Embeddings | 61.99 | **36.58** | **31.10** | 54.44 | **0.3599** | 12.42 |
| FleetAgent | Vector Embeddings | **93.26** | 33.24 | 27.52 | **55.93** | 0.3568 | **12.12** |
| Gemini-2.5-Flash | Language Description | 89.33 | 38.86 | 37.53 | 78.61 | 0.4478 | 16.00 |

In addition to the advantages of system-level evaluation, we also conduct experiments at the model performance level, highlighting the effectiveness of model design compared with general-purpose VLMs.

Our experimental results on the NU-EVAL dataset demonstrate the effectiveness of FLEE-TAGENT's vector embedding approach compared to other input modalities and multiple proprietary and open-sourced VLMs. We evaluate models using three categories of metrics: (1) **NLP metrics** including BLEU (B), ME-TEOR (M), and ROUGE (R) to assess linguistic quality and fluency of generated responses; (2) **context-aware evaluation** through Lingo-Judge (Marcu et al., 2023), evaluating the semantic appropriateness and contextual rele-

Table 3: Comparison on Nu-X dataset

| Model | Input Modality | B | M | R |
|---|---|---|---|---|
| GPT-4o | Raw Images | 2.45 | 10.5 | 23 |
| | Language Description | 14.35 | 11.93 | 5.85 |
| Gemini-2.5-Flash | BEV Images | 31.56 | 18.63 | 13.94 |
| | Language Description | 27.77 | 18.67 | 12.75 |
| Qwen-2.5VL-7B | BEV Images | 23.41 | 18.25 | 10.97 |
| | Language Description | 2.37 | 8.28 | 4.29 |
| TOD3Cap | - | 2.45 | 10.5 | 23 |
| HintAD | - | 4.18 | 13.2 | 27.6 |
| ALN-P3 | - | 5.59 | 14.7 | 35.2 |
| FleetAgent | Vector Embeddings | 76.96 | 19.51 | 26.72 |

vance of outputs in driving scenarios. The accuracy is calculated based on an acceptance threshold of 0.3; and (3) **task-specific performance** measured by the Intervention Failure Rate, which is calculated as the percentage of samples where the method fails to trigger intervention on cases

that need intervention, directly capturing the practical effectiveness of the model in the teleoperation scenario.

As shown in Table 2, FLEETAGENT achieves exceptional performance across these metrics, outperforming all baseline methods, including GPT-4o and Qwen-2.5VL-7B, across different input modalities (raw images, BEV images, and language descriptions). **Despite having the lowest communication and computational costs and the fastest response speed, FLEETAGENT achieves the lowest intervention failure rate of 12.12% and the highest score from Lingo-Judge, demonstrating superior task-specific performance crucial for real-world deployment.** The ablation study comparing FLEETAGENT with and without token prioritization demonstrates the effectiveness of the Context Tokens Prioritization Module. With a negligible performance difference, models equipped with the module **reduce the required number of tokens by 43.7%, resulting in a 14.8% acceleration**. Notably, the results on both GPT and Qwen confirm that a general-purpose VLM not pretrained on autonomous driving data can understand the situation via camera images and textual descriptions, but fails to work well on BEV images.

To make the performance comparable with existing VQA and driving explanation methods, we also experiment on the Nu-X dataset (Ding et al., 2024). As shown in Table 3, though FLEETAGENT was only trained on the Nu-X dataset while other methods use multiple datasets for grounding the MLLMs to the autonomous driving, our method can outperform the previous SoTA methods like ALN-P3 (Ma et al., 2025) and baseline methods.

### 6.3 QUALITATIVE RESULT

Fig. 4 provides an example to showcase FLEETAGENT's capability in understanding driving scenarios and providing intuitive explanations and evaluations for human teleoperators. Other methods, including Gemini-2.5-Flash, consider that this scenario needs to be taken over by a remote operator because it is attempting to halt on the lane divider. However, this is a safe and reasonable drive yielding to an oncoming vehicle. The context tokens prioritization helps in this case because irrelevant contexts are ignored, and interacting objects are preserved.

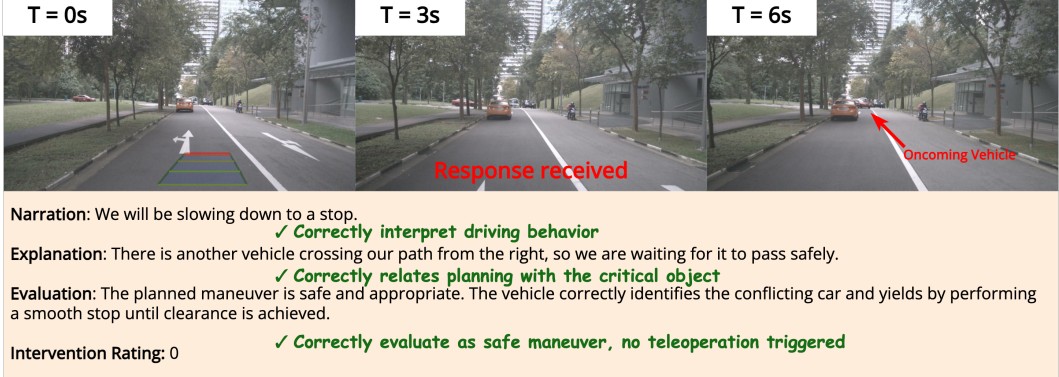

Figure 4: Qualitative Result

### 7 CONCLUSION

FLEETAGENT presents an alternative paradigm using on-cloud MLLM to evaluate the AV planning results, supplementing the current VLA and VQA paradigms for MLLMs' applications in autonomous driving, establishing a practical foundation for scalable, explainable teleoperation support systems. We achieve significant system-level improvements: a $625\times$ reduction in communication bandwidth and a $16.54\times$ reduction in cache size, without sacrificing task performance: obtaining a 41% improvement in BLEU score and an 11% reduction in intervention failure rates. Our Nu-Eval dataset will also provide a valuable resource for future research. The limitation lies in the supervision of the context tokens prioritization module during training, where no explicit labels for this relevance are available. As a future direction, we plan to conduct closed-loop validation and a user study on simulators and real-world vehicles.

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
