# OpenReview forum: "FleetAgent: Natural Language Driving Explanation and Evaluation for Vehicle Teleoperation"
_ICLR.cc/2026/Conference — Submitted to ICLR 2026_

### Official Review · Reviewer_bbDM · 2025-10-30

**Soundness:** 3
**Presentation:** 3
**Contribution:** 2
**Rating:** 4
**Confidence:** 3

**Summary:**

This paper presents FLEETAGENT, a cloud-assisted vision-language framework for fleet-level autonomous vehicle teleoperation. Instead of transmitting high-bandwidth sensor data, each vehicle uploads compact vectorized representations of its environment, including detected objects, dynamic trajectories, and planned maneuvers. A specialized encoder called VECFORMER converts these vectors into embeddings, which are then processed by a large VLM to generate natural-language explanations and intervention scores—deciding when human teleoperators should intervene. To improve efficiency, the system employs a differentiable top-K context selection mechanism (based on Gumbel-Softmax) that identifies the most relevant contextual vectors for reasoning. Experiments on NU-EVAL and VECEVAL datasets show that FLEETAGENT reduces bandwidth by 625×, cuts cache cost by 16.5×, and achieves strong interpretability and safety-intervention accuracy.

**Strengths:**

+ Innovative problem formulation: The paper defines a new and practically relevant task—language-based teleoperation reasoning from vectorized driving data—bridging large-scale autonomy and vision-language understanding.

+ Elegant and efficient system design: The combination of vectorized inputs + VECFORMER encoding + differentiable context selection enables both interpretability and communication efficiency, demonstrating real bandwidth and latency advantages.

+ Strong empirical validation and practicality: The authors provide extensive experiments on multiple datasets with concrete system-level metrics (bandwidth, cache, latency) and model-level metrics (BLEU, ROUGE, intervention accuracy), making the approach convincing for real-world deployment.

**Weaknesses:**

1. Limited gain in Intervention Failure Rate (IFR) and potential evaluation bias.
Although FLEETAGENT demonstrates strong system-level efficiency, its improvement in Intervention Failure Rate (IFR) appears modest. On the NU-EVAL benchmark, the reported IFR is 12.12%, only a 1.51-point absolute reduction compared with the GPT-4o baseline (13.63%), and within ~2–4 points of other strong baselines (16.05%, 15.90%, 15.24%). Such a small margin may fall within statistical noise, especially since no significance test, variance analysis, or sensitivity study (e.g., threshold vs. IFR curves) is reported. Moreover, because FLEETAGENT is fine-tuned on datasets highly similar to the evaluation sets, its responses naturally match the ground-truth style and template, which can inflate n-gram metrics (BLEU/ROUGE/ChrF) without necessarily improving explanatory quality or decision usefulness. In other words, the model may learn to mimic the format rather than produce substantively better reasoning. The evaluation should therefore include format-agnostic judging—e.g., LLM-as-a-judge and human-as-a-judge with blind protocols, report inter-rater agreement and confidence intervals, and add cross-template robustness (paraphrased GT, randomized phrasing) and unseen-domain tests to ensure gains are not merely artifacts of response-format alignment.

2. The paper exclusively adopts vectorized embeddings as the vehicle-to-cloud representation, omitting any version of FleetAgent trained directly on raw images. Could the authors clarify the rationale behind this design choice? Training a vision-language variant that processes raw images could leverage pretrained SOTA VLMs with stronger world knowledge transfer, richer scene semantics, and potentially lower training cost via lightweight fine-tuning instead of building a separate vector-based encoder. It would be helpful if the authors could discuss the trade-offs—e.g., bandwidth vs. accuracy, interpretability vs. scalability—and whether a hybrid approach (vector + vision) might further improve generalization and efficiency.

3. Qualitative results are too few, lacking failure cases and comparisons with other methods.

**Questions:**

see above

---

> ### Author Response · Authors · 2025-11-21
>
> Thank you for your thorough review of our paper. We appreciate your recognition of our contributions in problem setting and system design, and we will try to address your concerns below.
>
> **W1: Performance and Evaluation Bias**
>
> We acknowledge the reviewer’s concern about the modest improvement in Intervention Failure Rate. While the absolute reduction may appear small, we would like to highlight several important considerations:
>
> - **System-Level Trade-offs**: The IFR should be considered alongside our dramatic system efficiency gains (625$\times$ bandwidth reduction compared with raw image input, 16.54$\times$ cache reduction with language description input), especially when the stronger baselines (e.g., GPT ) are subject to service availability and quality.  Even maintaining comparable IFR while achieving these efficiency improvements represents a significant contribution for real-world deployment at scale.
>
> - **LLM-as-a-judge Evaluation:** We have included Lingo-Judge as one of the key statistics in the paper, where our model still performs the best across all tested methods. The question provided to Lingo-Judge does not specify any requirements for the response format or sentence pattern.
>
> **W2: Choice of V2N Representation**
>
> This is a good question about our design choices. First, we agree with the reviewer that the raw image can 1. Utilize existing pre-trained models. 2. Carry more semantic information. However, using vectorized representations is the starting point of our system design, for these reasons:
>
> - **Bandwidth Constraint**: For fleet-scale deployment with potentially thousands of vehicles, the 625× bandwidth reduction is essential. Raw image transmission would make the system impractical at scale, either considering the latency issue or the transmission cost. To achieve such a communication volume, using raw images is impractical.
>
> - **Interpretability and Deployment Constraint**: Vectorized representations provide explicit semantic structure (objects, trajectories, lanes) that enhances interpretability compared to end-to-end vision models. Additionally, vehicles with onboard planning capability already compute vectorized scene representations, requiring no additional on-board computing.

---

### Official Review · Reviewer_Ajwj · 2025-11-01

**Soundness:** 2
**Presentation:** 2
**Contribution:** 1
**Rating:** 4
**Confidence:** 5

**Summary:**

The paper proposes FleetAgent, a cloud-hosted multimodal LLM that consumes vectorized observations and planned trajectories to generate natural-language explanations. The approach centers on replacing tokenizers/vision encoders with a vector encoder (VecFormer) plus a differentiable top‑K context selection, and introduces a nuScenes-derived dataset with synthetic imperfect plans and NL annotations for explanation and evaluation.

**Strengths:**

1. This paper is generally readable and easy to understand.
2. This paper introduces a new dataset, NU-EVAL, which is extended from nuScenes.
3. The architectural and training pipeline is coherent and competently specified, and the quantitative results are consistent with expected trade-offs: vector inputs reduce bandwidth while maintaining competitive explanation and plan-evaluation quality.

**Weaknesses:**

1. The novelty of the architecture is limited. Designing a vector-to-LLM interface with attention-based token prioritization and Gumbel‑Softmax selection is an incremental adaptation of known components rather than a conceptually new modeling contribution.
2. The technical significance is not significant. The core improvements are largely architectural, packaging, and efficiency engineering around vectorized inputs to a pre-existing VLM, with the selection mechanism and training stages reflecting standard practices, and without introducing a fundamentally novel learning objective or evaluation framework.
3. The dataset contribution is not very strong. Many nuScenes-based VQA/explanation datasets already exist (e.g., NuScenes‑QA, NuInstruct, NuPrompt, NuPlanQA, DriveLM, Nu‑X), and re-deriving a plan‑explanation set on nuScenes with synthetic imperfect plans does not substantively expand coverage, realism, or scale for teleoperation safety assessment. Moreover, the nuScenes is a very small dataset.
4. Imperfect plans are synthesized by sampling from clustered human trajectories rather than sourced from actual autonomy failure logs, calling into question whether the evaluation meaningfully reflects the edge cases where teleop decisions matter most.
5. The paper acknowledges extensive VLM-for-driving work and enumerates multiple nuScenes/nuPlan-derived datasets for captioning, grounding, and reasoning. Given this landscape, positioning the contribution as a new paradigm is not fully convincing, as many elements, like the language explanations, plan reasoning, and nuScenes-based supervision, are already common.

**Questions:**

See weaknesses.

---

> ### Author Response · Authors · 2025-11-21
>
> Thank you for your review and constructive feedback. We want to address your concerns as follows:
>
> **W1 & W2: Limited novelty and technical significance**
>
> While individual components (transformer encoders, masked pretraining) are established, we introduce the task f(O, P) → (L, i), where we evaluate planning outputs rather than generate them, which is different from existing VQA or end-to-end planning approaches. The new problem setting requires the model to capture the relationship between trajectory planning and the surrounding environment, while the previous setting only required a thorough understanding of the surrounding environment. The need to capture this relationship motivates the design of the token selection module, where we tailored the Gumbel-Softmax mechanism for our application, enabling the end-to-end learning of context prioritization.
>
> By establishing baselines and datasets for cloud-based teleoperation support, we want to enable future research in this underexplored but practically important area.
>
> **W3: Dataset contribution**
>
> While nuScenes-based datasets exist, NU-EVAL uniquely provides:
>
> - **Plan evaluation labels** rather than just descriptions
> - **Paired perfect/imperfect trajectories** with scores indicating the urgency of intervention
>
> Existing datasets (NuScenes-QA, Nu-X, etc.) focus on scene understanding or action explanation, not planning evaluation for teleoperation triggering.
>
> **W4: Synthetic imperfect plannings**
>
> Our method was designed to generate physically plausible but incorrect trajectories efficiently at scale. We chose this approach because:
>
> - It ensures generated trajectories remain kinematically feasible (avoiding unrealistic discontinuities). At the same time, physically impossible trajectories can be easily filtered out and identified.
> - It covers fundamental decision errors (e.g., going straight when the vehicle should stop) that require intervention. Most failures occur before the decision-making and planning procedure (e.g., perception and planning), which usually lead to the above-mentioned decision errors.
> - It enables scalable dataset creation without manual annotation of each failure case.
>
> **W5: Positioning as a new paradigm**
>
> We clarify that the "new paradigm" refers specifically to the way MLLM is used in the autonomous driving domain. Existing work on this topic typically 1. uses MLLMs to achieve better scene understanding, and 2. uses MLLMs as teacher models to improve the performance of light-weight machine learning models. Our proposed FleetAgent now serves as an interface between the large-scale automated vehicle fleet and human teleoperators, selecting vehicles in need of intervention and providing intuitive explanations and evaluations on the scene.

---

### Official Review · Reviewer_UgRP · 2025-11-01

**Soundness:** 3
**Presentation:** 2
**Contribution:** 2
**Rating:** 4
**Confidence:** 4

**Summary:**

This paper introduces a cloud-based Multimodal Large Language Model framework, FLEETAGENT, designed to assess the driving plan and context of an Autonomous Vehicle (AV) to determine the necessity of remote human teleoperation. The paper tends to bridge large-scale driverless fleets and human teleoperators, and provid both evaluation and explanation to enable rapid situation awareness for remote drivers. It leverages vectorized representations involving many information to generate compact context embeddings via a custom VecFormer encoder. A new dataset designed for this new proposed task is also provided.

**Strengths:**

- This paper targets a underexplored yet (possibly) critical issue, scalable teleoperation support to address both efficiency and explainability for fleet-level autonomous driving.
- The proposed VecFormer encoder efficiently transforms structured driving context into embeddings suitable for LLM reasoning.
- The dataset extends nuScenes with imperfect trajectory generation and human-verified explanation annotations, and may fill an important gap between driving VQA and plan-assessment datasets.

**Weaknesses:**

- The framework focuses on explanation and plan evaluation but does not assess whether FleetAgent’s judgments lead to improved teleoperation decisions or reduced risk in real or simulated driving loops.
- FleetAgent mainly integrates known ideas—vector embedding, transformer encoding, masked pretraining and instruction tuning—into a teleoperation pipeline. Method novelty is very limited.
- Lines 74 said "actual limitaitons of deploying MLLM for real-world applications.." what are the limitations? Deployment from device to cloud?
- Better provide more demonstrations and user study or human preference evaluation to verify whether FleetAgent’s explanations are actually useful for teleoperators.
- Can we use this pipeline to different scenarios, apart from nuScenes-based ones.

**Questions:**

Please emphasize the significance and scientific contribution of the proposed setting within the context of AI and machine learning research, rather than primarily as a telecommunication or systems-engineering study. Explain any novel methodological designs and describe whether any closed-loop evaluations or real-time validation experiments can be conducted

---

> ### Author Response · Authors · 2025-11-21
>
> Thank you for your thorough review and constructive feedback.
>
> **Method Novelty and Scientific Contribution**
>
> While individual components (transformer encoders, masked pretraining) are established, we introduce the task f(O, P) → (L, i), where we evaluate planning outputs rather than generate them, which is different from existing VQA or end-to-end planning approaches. The new problem setting requires the model to capture the relationship between trajectory planning and the surrounding environment, while the previous setting only required a thorough understanding of the surrounding environment. The need to capture this relationship motivates the design of the token selection module, where we tailored the Gumbel-Softmax mechanism for our application, enabling the end-to-end learning of context prioritization.
>
> By establishing baselines and datasets for cloud-based teleoperation support, we want to enable future research in this underexplored but practically important area.
>
> **Responses to Specific Weaknesses**
>
> - **Limitation of MLLM Deployment for Teleoperation**: The limitations refer to: (1) computational constraints preventing onboard MLLM deployment, (2) bandwidth limitations for streaming raw sensor data at fleet scale, and (3) latency requirements for teleoperation. We will clarify this in the revision.
>
> - **Generalizability**: While demonstrated on nuScenes, our vector-based approach is dataset-agnostic. The VecFormer can process any structured driving representation (lanes, objects, trajectories). Notably, nuScenes covers scenarios from different areas (including the U.S. and Singapore) with different traffic rules. Results on our proposed VecEval dataset and the existing Nu-X dataset demonstrate the generalizability across different language patterns.
>
> - **Closed-loop validation & User Study**: Conducting closed-loop evaluation and a user study in both CARLA (based on the existing TeleCarla framework) and a real-world environment will be our next step.

---

### Official Review · Reviewer_d3mg · 2025-11-03

**Soundness:** 3
**Presentation:** 3
**Contribution:** 3
**Rating:** 6
**Confidence:** 3

**Summary:**

The authors proposed FleetAgent, a cloud-based Multimodal Large Language Model system designed to evaluate the driving decisions of autonomous vehicles and trigger remote human intervention when necessary. By utilizing a vectorized scene representation and a custom-built encoder called VecFormer, the system significantly reduces communication and computational overhead while generating natural language explanations and evaluations. The authors also construct the Nu-Eval dataset based on nuScenes, which includes imperfect driving behaviors and natural language annotations. Experiments demonstrate that FleetAgent outperforms existing methods in both system efficiency and task performance, offering a scalable and explainable solution for the teleoperation of large-scale autonomous vehicle fleets.

**Strengths:**

1.The system design is highly practical and scalable.The proposed FleetAgent system places computational tasks in the cloud, adding no onboard computational burden, while the vectorized input reduces communication demands, making it highly suitable for large-scale fleet deployment.

2. The authors designed VecFormer, a dedicated encoder for vectorized driving scenes that replaces traditional visual or text encoders. It effectively compresses input length while preserving key semantic information and introduces a differentiable Top-K selection mechanism to improve reasoning efficiency.

3.The authors constructed the Nu-Eval dataset, which includes both real and synthetic driving trajectories, natural language explanations, and intervention scores. A multi-stage training strategy effectively enhances the model's generalization capability with limited annotated data.

**Weaknesses:**

1.In Section 4.2.1, the generation of imperfect planning relies on random sampling from clustered trajectory categories.This method may not adequately capture the nuanced and context-specific errors that real-world autonomous systems make under complex or adversarial conditions.

2.In Section 5.2, the token prioritization mechanism in VecFormer depends on a manually set top-K value.While the Gumbel-Softmax sampling enables differentiable selection, the choice of K is heuristic and fixed across all scenarios. This may lead to under-selection in complex scenes or over-selection in simple ones, potentially affecting both model accuracy and inference speed.

3.In Section 6.1 and 6.2,the authors do not provide human-subject studies to assess whether FleetAgent’s natural-language explanations actually enhance teleoperator understanding.Also, the system-level experiments in Section 6.1 ignore network latency, bandwidth fluctuation, and other real teleoperation constraints.

**Questions:**

Please see the weaknesses above.

---

> ### Author Response · Authors · 2025-11-21
>
> Thank you for your thorough review and constructive feedback. We appreciate your recognition of our system's practical design and the VecFormer architecture. We address your concerns below:
>
> **W1: Imperfect Planning Generation Method**
>
> Our method was designed to generate physically plausible but incorrect trajectories efficiently at scale. We chose this approach because:
>
> - It ensures generated trajectories remain kinematically feasible (avoiding unrealistic discontinuities). At the same time, physically impossible trajectories can be easily filtered out and identified.
> - It covers fundamental decision errors (e.g., going straight when the vehicle should stop) that require intervention. Most failures occur before the decision-making and planning procedure (e.g., perception and planning), which usually lead to the above-mentioned decision errors.
> - It enables scalable dataset creation without manual annotation of each failure case
>
> We agree that incorporating more sophisticated error models (e.g., perception-induced errors, sensor degradation scenarios) would improve realism in terms of error distribution. We will enhance this in future work by incorporating failure patterns from real-world logs.
>
> **W2: Fixed Top-K Selection in VecFormer**
>
> Thanks for raising the point about the fixed K value. The current selection of K strikes a balance between context length and performance. We acknowledge that an adaptive context selection mechanism may improve the performance, and the current design already provides the foundation for this extension, requiring primarily modular modifications rather than fundamental changes.
>
> **W3: Network Constraints and User Studies**
>
> Our motivation is grounded in the network constraints. Compared with other message types, the vectorized representation is the one posing the least network demand, reducing the communication demand by up to 625 times.
>
> A user study will be an important way to assess the gains provided by natural language explanation and evaluation, and will be our next step to extend the work.

---

### Meta-Review · Area_Chair_L4cM · 2026-01-10

**Summary:**

This work proposed FleetAgent, a cloud-based Multimodal Large Language Model system designed to evaluate the driving decisions of autonomous vehicles and trigger remote human intervention when necessary. Experiments demonstrate that FleetAgent outperforms existing methods in both system efficiency and task performance, offering a scalable and explainable solution for the teleoperation of large-scale autonomous vehicle fleets.

**Reviewer Concerns:**

The reviewer concerns are summarized as follows:

- Imperfect Planning Generation Method
- Fixed Top-K Selection in VecFormer
- Network Constraints and User Studies. Reviewer request the "network latency, bandwidth fluctuation, and other real teleoperation constraints." Authors do not reply with detailed results and discussions.
- Whether FleetAgent’s judgments lead to improved teleoperation decisions or reduced risk in real or simulated driving loops.
- Method novelty is very limited. by several reviewers.
- Better provide more demonstrations and user study or human preference evaluation to verify whether FleetAgent’s explanations are actually useful for teleoperators.
- The technical significance is not significant.
- The dataset contribution is not very strong.
- Limited gain in Intervention Failure Rate (IFR) and potential evaluation bias.
- Qualitative results are too few, lacking failure cases and comparisons with other methods.

**Reviewer Scores:**

Authors only respond partical concerns raised by reviewers. A lot of review comments are not anwersed. This is a clear signal that the current manuscript is not ready for publication at ICLR.

---

### Decision · Program_Chairs · 2026-01-26

Reject